# POPULATE-A-SCENE: AFFORDANCE-AWARE HUMAN VIDEO GENERATION

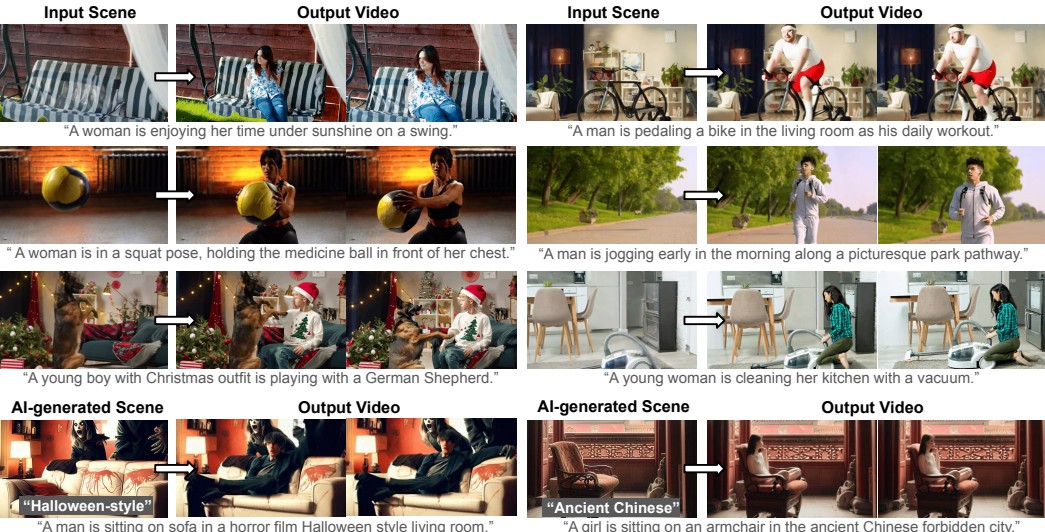

Figure 1: We repurpose a text-to-video generation model as a human-world interaction simulator. Given a scene image and a prompt, our model inserts a person into the environment and generates a video of them naturally interacting with the scene. The scene can be real images (top) or synthesized by image generative models (bottom). Notably, there is no need for any mask, location bounding boxes, or pose sequences to guide the human insertion – our method takes care of affordance-aware human movement prediction entirely within the video model.

## ABSTRACT

Can a video generation model be repurposed as an interactive world simulator? We explore the affordance perception potential of text-to-video models by teaching them to predict human-environment interaction. Given a scene image and a prompt describing human actions, we fine-tune the model to insert a person into the scene, ensuring coherent behavior, appearance, visual harmony, and scene affordance. Unlike prior work, we infer human affordance for video generation (i.e., where to insert a person and how they should behave) from a single scene image, without explicit conditions like bounding boxes or body poses. An in-depth study of cross-attention heatmaps demonstrates that we can uncover the inherent affordance perception of a pre-trained video model without labeled affordance datasets.

## 1 INTRODUCTION

Scaling data, compute, and model parameters in video generation models presents a promising avenue for developing highly capable simulators that can accurately replicate complex physical worlds (Polyak et al., 2024; Brooks et al., 2024b), complete with diverse objects and people that interact and coexist within them. Nevertheless, humans are not merely passive observers, but rather active participants in the world. Human understanding of affordance (Koffka, 1999; Gibson, 1996; Norman, 2013) enables purposeful engagement with surroundings and adaptive behavior by recognizing potential actions afforded by an object's physical properties. It remains unclear whether video generation models can interpret and replicate intricate semantic aspects of the world, such as contextual understanding and dynamic behavior, beyond the capabilities of traditional graphics pipelines.

Affordance, or "opportunities for interaction" (Gibson, 1996), has inspired extensive research in vision and psychology. Traditional affordance prediction relies on data-driven approaches using 3D information (Hassan et al., 2021), specifically labeled datasets (Wang et al., 2017; Gupta et al., 2011; Fouhey et al., 2012; Delaitre et al., 2012; Chen et al., 2023b), or one-shot large foundational models (Li et al., 2024). However, these methods rely on domain-specific annotations, which are challenging to obtain. In contrast, recent advancements in generative models offer the potential to create realistic human-scene media content using vast amounts of in-the-wild media data. Kulal et al. (2023), for example, predicts a human's pose and appearance in a scene but is restricted to static images with a given position mask.

In this work, we demonstrate that throughout the intricate process of video generation, the model learns to generate human activities and motions that adhere to the affordance constraints dictated by the physical world. To better study affordance modeling, we propose augmenting a pre-trained text-to-video model (Polyak et al., 2024) with an additional scene conditioning branch. This modification formulates the problem as a conditional video generation task: given a scene represented by an image, the model is tasked with introducing natural human motion and interactions to the scene. We discover that pre-trained video generation models can rapidly adapt to this new task by fine-tuning on a relatively small-scale scene conditioning dataset. We then validate the affordance perception abilities through an extensive study of the cross-attention feature heatmaps, a key module that enables the model to follow language prompts.

Unlike prior work, our model does not require input masks, bounding boxes, or pose sequences to specify regions or patterns of human behavior, which makes it an interaction *simulator* that reasons about semantics and affordance properties in the scene, instead of merely a human *renderer* that turns given pose signals into pixels. During inference, the model can process a wide array of environment-action combinations to generate diverse interactive videos, not limited to interaction with the single, salient object in a complicated scene. Fig. 1 demonstrates results of our model without aggressive cherry-picking. In particular, the last row of Fig. 1 illustrates a "movie studio" pipeline where input scenes are generated using a text-to-image model (Dai et al., 2023), and our model seamlessly integrates actors into these scenes without requiring 3D capture. Our results lower the barrier for amateur AI video creators by eliminating the need for explicit body pose signals, as they are required in most AI human video models but challenging to synthesize.

In short, this work makes the following contributions:

- We address affordance-aware human video generation, where we generate video of subject(s) interacting with a given environment image, *without* telling the model where the subject(s) are and how their poses look like.

- We apply the dual-stream conditioning mechanism with a minimal grounding module to model affordance, and thus reveal the affordance capabilities of video generation models through in-depth analysis.

- We demonstrate our model's ability to generalize across diverse environments and actions through a synthetic benchmark created with vision-language models.

## 2 RELATED WORK

**Text-to-video generation.** Text-to-video generation synthesizes plausible, temporally coherent, and condition-aligned videos from textual prompts. Recent models show rapid progress (Ho et al., 2022; Singer et al., 2022; Ge et al., 2024; Blattmann et al., 2023; Brooks et al., 2024a; Polyak et al., 2024; Wang et al., 2023a; Bar-Tal et al., 2024; Chen et al., 2024; Esser et al., 2023; He et al., 2022), with growing interest in replacing U-Nets by Transformer architectures (Vaswani, 2017; Gupta et al., 2023; Ma et al., 2024; Brooks et al., 2024a; Polyak et al., 2024), inspired by DiT (Peebles & Xie, 2022). We build on the Transformer-based MovieGen (Polyak et al., 2024), fine-tuning it for human–scene interaction affordances. Some approaches augment generation with an input image frame to guide motion (Zeng et al., 2023; Gong et al., 2024; Ren et al., 2024); in contrast, we condition on an empty scene image that serves only as a "playground" for population but not appear directly in the video. Beyond building a creative application, we study the intermediate cross-attention maps in diffusion models, known to capture meaningful token–pixel correspondences Hertz et al. (2023); Chefer et al. (2023); Wen et al. (2025), to probe affordance perception.

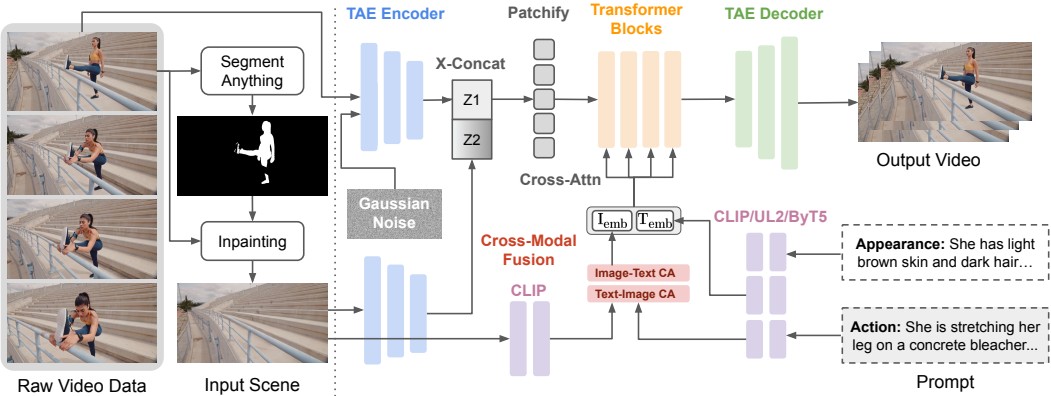

Figure 2: We start by removing humans from raw frames to create synthetic empty-scene and human-video data pairs. We employ a dual-conditioning mechanism, using channel concatenation and cross-attention, to condition the T2V model on an additional scene image. We design a fusion module to facilitate interactions between image and action-text features while locating the desired action position. The fine-tuning pipeline trains a Transformer architecture based on flow matching.

**Human video generation.** Human video generation evolves alongside rapidly advancing generic video generative models. Generating realistic human content is inherently challenging due to complex body topology, strong priors on interaction plausibility, and audiences' sensitivity to even minor artifacts. Existing methods use motion guidance to improve video faithfulness, leveraging signals such as OpenPose (Hu et al., 2023; Wang et al., 2023b), DensePose (Xu et al., 2024; Karras et al., 2023), SMPL (Zhu et al., 2024), or a driving video (Yatim et al., 2024). These works focus on human video generation with the subject as the sole salient element, without modeling human-environment interaction. Our work differs in that we reason about natural human-scene interaction without compromising human quality. Notably, our method requires no auxiliary conditions such as position bounding boxes (Singh et al., 2023; Kulal et al., 2023) or motion sequences, relying instead on the internal affordance inference potential of video models.

**Human-scene interaction modeling.** A fundamental task in human-environment modeling is motion prediction in 3D scenes (Li et al., 2019; Li & Dai, 2024; Wang et al., 2024; Kim et al., 2025). Related work in 2D explores interaction image and video generation from a scene, mostly using some location or body pose signals as conditions (Ostrek et al., 2023; Saini et al., 2024; Yang et al., 2024a; Hu et al., 2025). Kulal et al. (2023) and Cao et al. (2025) claim to predict affordance by inserting a subject into a static scene, but require a bounding box indicating the position. Shan et al. (2023) inserts moving humans into a street scene, but restrict actions to predefined walking motions. Singh et al. (2023) predict fine-grained masks for insertion based on scene and text but do not explicitly model environment interaction. Jin et al. (2025) builds on similar ideas as ours, but focuses on static images with non-human objects, which lack intricate interactive dynamic behaviors. Our work instead requires no semantic priors for where and how human-scene interaction occurs.

**Affordance.** Psychologist J.J. Gibson defines *affordance* as the possibilities an environment offers an individual (Gibson, 1996; Norman, 2013) and views affordance perception as essential to socialization. Following cognitive psychology, computer vision research explores scene and object affordance prediction (Chuang et al., 2018; Tang et al., 2023) and affordance learning from observing human-scene interactions (Delaitre et al., 2012; Fouhey et al., 2012; Wang et al., 2017; Bahl et al., 2023; Chen et al., 2023a). Inspired by these discussions, we study how generative models perceive affordance by not passively watching but actively *creating* interactive videos.

## 3  PRELIMINARY: TEXT-TO-VIDEO GENERATION

In this work, we leverage Movie Gen (Polyak et al., 2024) as our base text-to-video model. Due to resource limitations, we conduct our experiments on a 4B-parameter model that generates 128-frame 256p videos as a proof of concept, instead of training the official 30B-parameter model that operates at 1080p. We highlight key architectural and training aspects incorporated into our experiments in the following section. Please refer to the supplementary material for more details.

**Temporal autoencoder.** Our model encodes RGB videos and images into a learned spatiotemporally compressed latent space using a Temporal Autoencoder (TAE) and generates videos in this space. The TAE encoder is designed by inflating the image autoencoders in Rombach et al. (2021), adding a 1D temporal convolution after each 2D spatial convolution and a 1D temporal attention after each spatial attention.

**Video generation backbone.** The model generates videos within a learned latent space as defined by the TAEs. The latent video code is segmented into patches via a 3D convolutional layer (Dosovitskiy, 2020), then flattened into a 1D sequence as input to the generation backbone. The backbone consists of Transformer (Vaswani, 2017) blocks with cross-attention modules inserted between self-attention and feed-forward networks, enabling text conditioning via text prompt embeddings. The model employs UL2 (Tay et al., 2023), ByT5 (Xue et al., 2022), and Long-prompt MetaCLIP (Xu et al., 2023) as text encoders, enabling semantic- and character-level text understanding.

**Flow matching.** The model is trained with Flow Matching (Lipman et al., 2023; Boffi et al., 2024), which iteratively transforms a prior Gaussian distribution into a sample from the target data distribution. During inference, an ordinary differential equation (ODE) solver transforms random noise into video latents. We use this training and inference framework for all experiments.

## 4 AFFORDANCE-AWARE VIDEO GENERATION

Our full pipeline is illustrated in Fig. 2. We define the problem in Sec. 4.1, explain the data processing procedure in Sec. 4.2, and illustrate the model architecture in Sec. 4.3.

### 4.1 TASK DEFINITION

Let $I$ be an image of a static scene, and let $T_h$ and $T_a$ be text prompts describing a human's appearance and action. We generate a video $V$ that depicts the given scene $I$ with an inserted human matching the appearance described by $T_h$ and performing the action in $T_a$. During training and inference, we provide no explicit guidance for the human's position or pose in the scene, allowing the generative model full freedom to position the action, simulate body movements, and render the video. Note that this is not image animation; the scene image serves only as a reference for the background appearance and the presence of semantically meaningful objects. We do not require the image to appear as a frame in the video, nor do we treat the scene as a static background for pasting the human without environmental animations or camera viewpoint changes.

### 4.2 TRAINING DATA

We explain our full data processing pipeline below. Representative data samples are shown in Fig. 3.

**Human filtering.** We curate our dataset by selecting human-related videos from the ShutterStock (Shutterstock, 2025) text-video dataset. We apply human detection to each middle video frame and retain only those with one or two detected persons. This filtering leaves us with around 250,000 videos with one person, and another approximately 171,000 videos with two people.

**Full body filtering.** We apply OpenPose (Cao et al., 2019) to videos that pass the previous stage, retaining those where knee keypoints are visible or face height and width fall below a threshold to avoid half-body or close-up shots.

**Pure background filtering.** We compute the color variance of background pixels in the middle frame of each video, retaining only those exceeding a threshold of 200. We also scan video captions and exclude those containing keywords like "a pure green background." This helps eliminate studio-recorded videos that lack background interaction.

**Human removal.** We take the first and last frames of each video, with GroundingDINO (Liu et al., 2023) detecting the central human subject and language-guided SAM (Kirillov et al., 2023) segmenting the human mask. We dilate the mask by 50 pixels to fully cover the human region and apply a text-to-image inpainting model with the negative prompt "human" for removal. For two-person videos, we remove one person at a time, creating two data samples from a single video. This results in a training dataset of (`text`, `image`, `video`) tuples representing (`action`,

`scene, interaction)`, including 217,530 samples for single-person data and 29,700 for two-person data. We handpick 300 samples per category for validation and detail the post-processing steps for the synthetic validation benchmark in Sec. 6.1.

**Prompt rewriting.** We use LLaMA 3 (Dubey et al., 2024) to rewrite video captions, separating out human-related prompts ($T_a$ and $T_h$) and removing sentences that pertain solely to the background. This allows the model to learn background information purely from the visual modality rather than text, promoting multimodal information fusion.

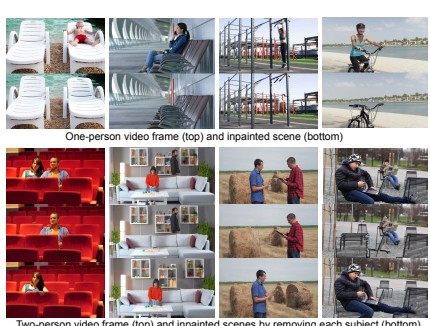

Figure 3: Examples in our dataset. Top/bottom row shows single/double-person data. Within each row, the top/bottom figures present the video frame before/after human removal.

Figure 4: Cross-attention maps of the video models. Top half is the pre-trained model where the presented scene is generated by the model, and bottom half is our scene-conditioned model with a real image as input. Attention is averaged across timesteps.

### 4.3 CONDITIONING MECHANISM

During fine-tuning, we aim to keep the original structure as much as possible, while exploring conditioning strategies to unfold a text-to-video model's innate ability to perceive affordance from a scene image. We discuss key strategies to condition the model on an additional image input.

**Masked latent concatenation.** To preserve background consistency with the given image $I$, we concatenate its latent $Z_1$ with the noisy video latent $Z_2$ along the channel dimension before feeding them into the Transformer backbone. Unlike image animation, our formulation permits environmental updates driven by the action prompt $T_a$ and natural camera effects. To balance control and flexibility, we progressively inject Gaussian noise into $Z_2$ with a temporal factor $\gamma_t$, which decays over time so that the scene initially aligns with $I$ but gradually allows modifications. This process can be written as:

$$\tilde{Z}_t = \text{Concat}(Z_1, (1 - \gamma_t)Z_2 + \gamma_t \epsilon), \quad \epsilon \sim \mathcal{N}(0, I). \tag{1}$$

**Fused text-image feature enhancer.** We augment the cross-attention conditioning branch with a fusion module that performs mutual attention between image and action-text embeddings, inspired by Liu et al. (2023); Li* et al. (2022). Following the base model, we concatenate three text embeddings (ByT5, UL2, MetaCLIP) into a unified textual representation, and extract a spatial-aware embedding from the CLIP image feature map. We apply deformable self-attention (Xia et al., 2023) to image features and standard self-attention to text features, followed by cross-modal alignment through separate image-to-text and text-to-image cross-attention. The fused representation is then concatenated with textual embeddings and injected into each Transformer block. Formally:

$$H = \text{Concat}(E_{\text{text}}, f_{\text{fuse}}(E_{\text{text}}, E_{\text{img}})), \tag{2}$$

where $E_{\text{text}}$ and $E_{\text{img}}$ denote text and image embeddings, and $f_{\text{fuse}}$ is the cross-modal fusion module.

**Controlled guidance scale.** Following the practice of InstructPix2Pix (Brooks et al., 2023), we leverage a controlled multi-scale guidance mechanism to control the strength of background scene image and action prompt. A higher image strength preserves scene consistency, while a higher text strength emphasizes human action and promotes plausible environmental updates. Training with dummy condition images helps maintain the pre-trained model's text-to-video capability and prevents overfitting to a specific dataset domain.

## 5 Unveiling Implicit Affordance Capability

We comprehensively analyze the implicit affordance modeling capabilities of our proposed model. In Sec. 5.1 we justify that affordance perceiving information can be unveiled by investigating the cross-attention modules, specifically which processes and regulates the CLIP text conditions. In Sec. 5.2 we apply our model on a real-world affordance prediction dataset. The primary objective of cross-attention is to select appropriate values $\mathbf{V}$ using the attention scores $\mathbf{S}$ determined by

$$\mathbf{S} = \text{softmax}(\mathbf{Q}\mathbf{K}^T/\sqrt{d}) \in \mathbb{R}^{n \times m} \tag{3}$$

Here, $\mathbf{Q} \in \mathbb{R}^{n \times d}$ represents the projected and flattened intermediate diffusion features. $\mathbf{K} \in \mathbb{R}^{m \times d}$ and $\mathbf{V} \in \mathbb{R}^{m \times d}$ are the projected features of the input text embedding. The attention map $\mathbf{S}$ provides a physical interpretation where each entry $(i, j)$ indicates the saliency of interaction between a spatial location $i$ and a token $j$ in the prompt. This saliency reflects how strongly a particular spatial feature is associated with a specific word, guiding the model in generating contextually relevant outputs.

### 5.1 Predicting Affordances via Cross-Attention

We explore the implicit affordance reasoning capability of video models by visualizing the $j$-th entry of the attention map $\mathbf{S}$ where the $j$-th token corresponds to an action-related term in the prompt. For example, given the input prompt "a woman holding the rope and riding a horse", we focus on visualizing the attention heatmap associated with the verb "holding" and "riding".

The top half of Fig. 4 shows the attention scores of the pre-trained T2V model. Trained exclusively on text-video pairs, the model exhibits a reasonable ability to perceive affordances while generating high-quality, faithful content. The heatmaps align well with action regions, highlighting the model's ability to associate generated spatial features with actions.

Building on this observation, we propose that conditioning the model on an additional scene enables it to perceive affordances in a *given, real* image. The bottom half of the figure shows that the model accurately identifies action locations in input images and the specific environmental elements involved in the interaction. Our heatmaps reveal internal affordance knowledge, capturing interaction opportunities in real images rather than merely serving as intermediate by-products during the process of synthetic content generation.

### 5.2 Real-World Affordance Prediction Experiment

We subsequently analyze our model's affordance perception using classical 2D affordance detection datasets. We filter the Purpose-Driven Affordance (PAD) dataset (Luo et al., 2021), retaining only images with no person and action verb-object pairs representing human actions (e.g., push, hit) and discarding passive object verbs (e.g., contain). This leaves us with 24 action verb categories, totaling 235 images with corresponding affordance masks. We create the prompts based on the affordance verb with LLaMA (Dubey et al., 2024), and pass in the image and prompts as inputs for our model.

In Fig. 5, we present heatmap visualizations, derived similarly to those in Sec. 5.1. We also compute the spatial accuracy (defined as pixel-wise IoU) between the binarized attention map and the ground-truth affordance mask across different layers and diffusion inference steps. We observe slightly higher scores in the initial layers, likely because the model processes semantic information early in generation. Even in the early steps, our model consistently predicts affordance through attention features. Accuracy decreases in later steps as the model shifts from perceiving high-level semantics to refining details of generated content. Peaks in the attention heatmap gradually transition from interaction regions to human content. The spatial alignment of the heatmap and ground-truth affordance maps highlights the video model's ability to perceive affordance in alignment with real-world data. Our model even outperforms the ground-truth by identifying the specific object parts relevant to each action. For example, it identifies the seat of a bench where people sit, rather than its legs.

## 6 Results

We present quantitative and qualitative results of our proposed affordance-aware human video generation models. Our model effectively generalizes across a variety of environments and actions, producing realistic human-scene interactions that adhere to affordance principles.

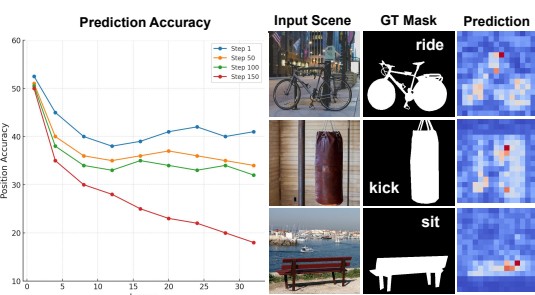 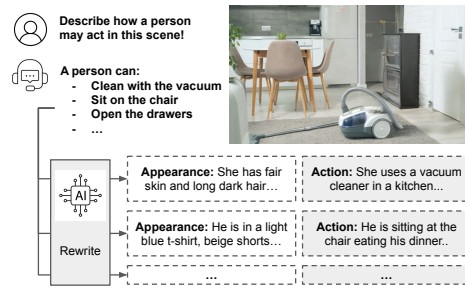

Figure 5: Affordance position accuracy across different steps and layers on a subset of PAD. The attention scores indicate strong predictive ability, and visualizations show that our model accurately locates detailed affordance information.

Figure 6: The synthetic action descriptions generated through our prompting process. We use a vision language AI agent to decide plausible actions in a scene, and rewrite the action into prompts.

## 6.1 EVALUATION DATASET

We aim to generate *diverse* actions interacting with more than one part of the environment, even within a fixed scene. We thus curate synthetic prompt sets based on real scene images. Specifically, we use a pre-trained vision-language model to generate two prompts per scene by asking, "What might a person do in this scene?" This process yields an evaluation set of 300 images, each paired with one original and two synthetic prompts. These prompts emphasize different objects or positions within a complex environment, allowing us to assess whether our model's generative ability extends beyond central, salient objects. We repeat this process for two-person scenarios, prompting interactions with both the scene and the existing person. Fig. 6 illustrates our benchmark pipeline.

## 6.2 BASELINES AND ABLATION

**Baselines.** To the best of our knowledge, there is no existing work on generating human videos in a scene without location or pose control. We therefore compare our methods with generic image/video editing and image-to-video solutions not tailored for humans. We compare with three image-based models: (1) **InstructPix2Pix** (Brooks et al., 2023) where we directly apply an image editing model on the empty scene image with the prompts. (2) **Flux Editing** which trains instructional image editing on Flux (Labs et al., 2025). (3) **Flux Inpainting** where we provide a groundtruth human mask as the inpainting position. We then compare with instruction-based video editing method (4) **AnyV2V** (Ku et al., 2024) where the scene is repeated for 2 seconds to a video, and then edited based on a prompt. We additionally compare with one open-source and three commercial video generation models (5) **CogVideoX** (Yang et al., 2024b), (6) **Runway Gen-4** (Runway, 2025) and (7) **Luma AI Ray-2** (Luma, 2024) where we apply image-to-video on the scene with the human action prompt. For (1), (2), (3), we attach a CogVideoX image animation model to the image results, exploring their potential to generate interaction videos in the same setting as ours. Note that we only do visual comparison and human evaluation on (6) and (7) without quantitative metrics, since free APIs are not publicly available for those commercial models.

**Ablation studies.** We compare with alternative designs of our model that remove key features, including latent concatenation, fused cross-attention, and Gaussian noise decay. Due to space limits, parts of the ablation results are presented in the supplementary material.

## 6.3 QUALITATIVE EVALUATION

**Human-scene interaction.** Fig. 1 presents inserting a human into a scene based on an action prompt. Fig. 7 shows adding a subject to interact with both the scene and an existing person who is considered a part of the scene. The model maintains pixel-level scene consistency while placing the subject correctly without a predefined mask. The generated video features natural camera movements, object updates in response to human actions, and scene animations.

**Diverse affordance.** In scenes with complex layouts and multiple interaction possibilities, our model inserts subjects while accounting for diverse scene elements and action-affording objects.

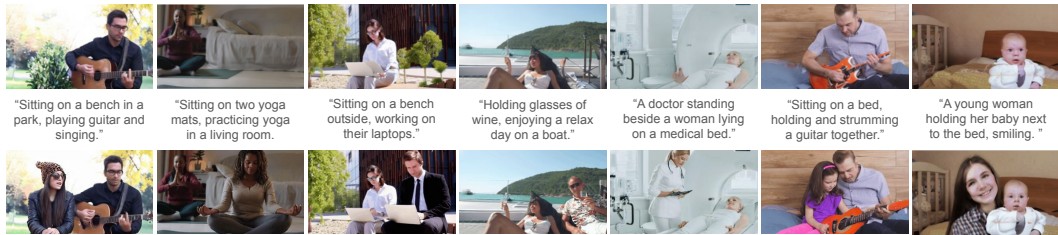

Figure 7: Our model is able to add an extra subject to a scene that contains one person. Here we consider the existing person as an organic part of the environment, and are able to synthesize interactions respecting both the background and the human in the scene.

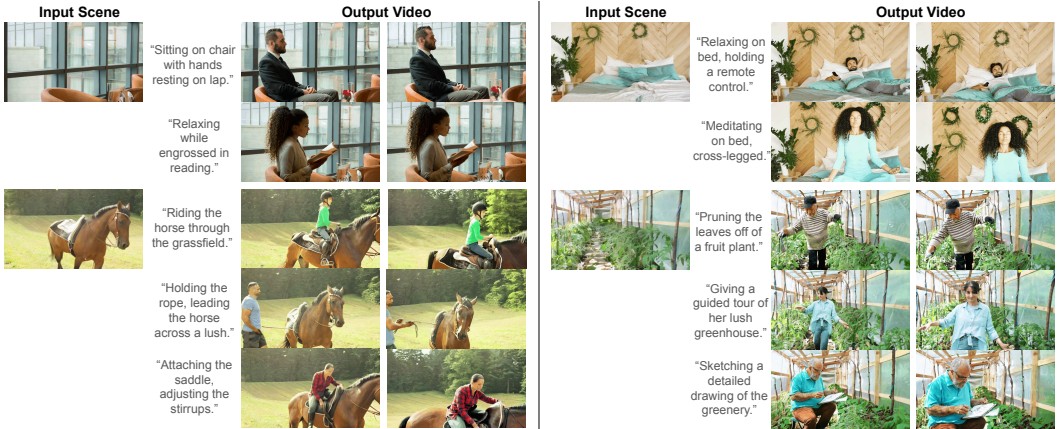

Figure 8: Our model generates diverse videos with multiple action prompts given the same scene. It identifies the correct way for an inserted subject to interact with the scene, and infers location, pose, action, spatial relationship without a pre-defined human mask prior.

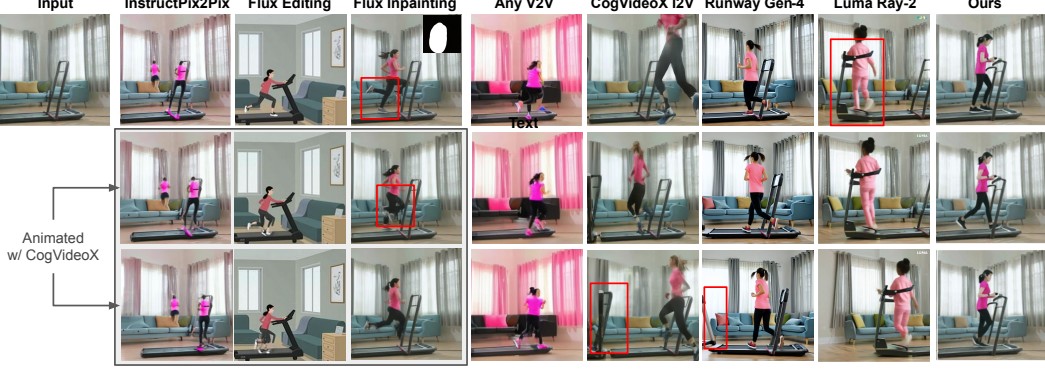

Figure 9: Comparison with baseline methods. Three rows are the first, middle and last frames for each method. The left three columns' models edit a static frame and animate it. The next four edit video directly. Note that Flux (Labs et al., 2025) Inpainting requires a user-defined mask as input, which eases the task and greatly assists the model in predicting human position. Yet, our model outperforms baselines in terms of human placing, motion simulation and appearance rendering. See video results and more comparison in the supplementary materials.

Fig. 8 illustrates how our model determines subject placement and imagines body poses for different actions (e.g., riding vs. standing beside a horse).

**Baseline comparison.** Fig. 9 demonstrates that our model achieves the highest semantic alignment and visual fidelity. Instruction editing methods like InstructPix2Pix and AnyV2V generate distorted human bodies and misattribute prompt concepts (e.g., applying "pink" to the treadmill or curtain

Table 1: Quantitative evaluation shows our method consistently outperforms baselines and alternative model design choices in visual quality, text alignment, and action faithfulness.

| Model | CLIP ↑ | FVD ↓ | Action ↑ |
|---|---|---|---|
| InstructPix2Pix | 0.19 | 302 | 0.14 |
| Flux Inpainting | 0.40 | 174 | 0.65 |
| Flux Editing | 0.23 | 332 | 0.63 |
| AnyV2V | 0.23 | 290 | 0.33 |
| CogVideoX | 0.38 | 199 | 0.69 |
| w/o x-concat | 0.46 | 185 | 0.76 |
| w/o cross-attn | 0.59 | 220 | 0.55 |
| w/o fusion | 0.65 | 171 | 0.85 |
| Ours | **0.67** | **168** | **0.88** |

Table 2: Human evaluation preference comparison with baselines and alternative designs.

| Model | SC (%) | HQ (%) | PA (%) | AP (%) |
|---|---|---|---|---|
| InstructPix2Pix | 100 | 98 | 100 | 96 |
| Flux Editing | 87 | 94 | 99 | 97 |
| Flux Inpainting | 95 | 79 | 60 | 57 |
| AnyV2V | 100 | 100 | 100 | 98 |
| CogVideoX | 68 | 87 | 74 | 89 |
| Runway Gen-4 | 54 | 65 | 67 | 70 |
| Luma Ray-2 | 55 | 59 | 69 | 75 |
| w/o x-concat | 99 | 48 | 53 | 76 |
| w/o cross-attn | 73 | 89 | 61 | 69 |
| w/o fusion | 54 | 52 | 58 | 60 |
| w/o noise decay | 76 | 48 | 56 | 53 |

instead of clothing). Editing methods based on Flux do not preserve scene styles and generate cartoon videos. Flux Inpainting distorts human bodies even when provided with an additional mask and fails to preserve pixel details in masked background regions (the yellow pillow disappears). Current best open-sourced and commercial image-to-video models like CogvideoX, Runway Gen-4, Luma Ray-2 all misinterpret the treadmill's affordance, place subjects in the wrong direction, and even hallucinate another treadmill on the left. Our models stand out by successfully preserving the background and simulating natural interactions between the subject and the treadmill.

## 6.4 QUANTITATIVE EVALUATION

We evaluate our model based on human video faithfulness, text-video alignment, and action quality. This corresponds to three major quantitative metrics: (i) **FVD** (Unterthiner et al., 2018), which quantifies the similarity between real and synthetic video embedding distributions. (ii) **CLIP** (Radford et al., 2021) similarity, which computes the average embedding similarity between the input prompt and each generated frame to assess prompt alignment. (iii) **Action Score**, computed by querying a pre-trained VQA model (Zhang et al., 2024) with "What action is the person performing in this video?" and measuring the CLIP similarity between the recognized motion and the ground-truth action prompt. The Action Score helps isolate interaction quality from the influence of appearance. For image-only baselines, we compute metrics on the animated video sequence using CogVideoX.

We quantitatively compare our model with baselines and ablated variants. Results in Tab. 1 show that our model consistently outperforms others in visual quality, text alignment, and action faithfulness.

## 6.5 HUMAN EVALUATION

We supplement our analysis with a structured A/B test human evaluation. We assess the results based on four criteria: **(i) Scene consistency (SC)** evaluates how well the video preserves the original scene, even with flexible camera angles and scene motions. **(ii) Human quality (HQ)** assesses the realism of the generated human body. **(iii) Text-prompt alignment (PA)** evaluates how accurately the generated action and appearance match the given prompt. **(iv) Affordance prediction (AP)** assesses the subject-scene interaction plausibility. Tab. 2 presents the percentage of subjects preferring each model, demonstrating that our model is consistently perceived as more realistic, natural, and capable of producing reasonable interactions compared to baselines and ablations.

## 7 CONCLUSION

We explore the ability of text-to-video models to perceive affordance and reason about interaction through the task of populating scenes with moving humans. Beyond serving them as a creative application, we show that video generative models implicitly learn affordance and can simulate affordance-aware activities through extensive analysis of attention features. We provide preliminary insights into effectively leveraging video generative models beyond appearance rendering, toward interaction simulation.

**Reproducibility Statement.** Due to copyright restrictions, we cannot release the training codebase or dataset. However, detailed descriptions of the base models in Polyak et al. (2024), together with the extensive implementation details provided in this paper, should give readers a clear understanding of our model architecture and training paradigm. We also include step-by-step documentation of our dataset processing pipeline in Sec. 4.2 and Sec. 6.1, which we hope will be useful for those building data curation pipelines on public-domain datasets. Moreover, our aim is not to present the strongest human video generation model, but rather to study how pre-trained T2V models perceive affordance from visual signals. The proposed conditioning mechanism and cross-attention analysis are broadly applicable to open-source models, and our results serve as a proof-of-concept intended to encourage further exploration. Upon acceptance, we will release the benchmark evaluation dataset we collected to enable fair comparisons in future work.

**Ethics Statement.** This work uses video data licensed from Shutterstock, with copyright purchased by the collaborating organization. The dataset was curated under proper usage rights and filtered to remove personally identifiable information and sensitive content. No private or unauthorized data sources were used. Our contributions are methodological and focus on affordance-aware human–scene video generation. While generative video models may raise concerns about potential misuse, our intention is to advance scientific understanding, and we emphasize that responsible deployment and adherence to copyright and ethical standards are essential.

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

# APPENDIX

## A   VIDEO RESULTS

We present the video version of all the data and results we show in the paper, along with additional results, to demonstrate the generalizability of our model. Please refer to the video folder for the results. You can also click on `video_results.html` link to open it with your favorite browser (loading faster in Chrome than Safari!) to see everything all at once. Specifically, we present results of the following kinds:

- Single-person insertion results.
- Two-person insertion results.
- Multi-prompt interaction results.
- Comparison with image-to-video baselines.

We hope those real video results can showcase the quality of our generative model. Note that we tried to not do aggressive cherry picking on those results. All of the shown videos are generated in one pass without tweaking the random seed, and picked out of around one hundred validation samples to cover a diverse range of interesting behavior.

## B   DATA PROCESSING DETAILS

### B.1   DATA FILTERING

We get the raw human-related dataset following the practice of video personalization in (Polyak et al., 2024). Specifically, we first get human videos by selecting videos with human-related concepts in their captions. We extract frames at one-second intervals and apply a face detector to keep videos that contain a single face and where the ArcFace cosine similarity score (Deng et al., 2019) between consecutive frames exceeds 0.5. This processing provides us with around one million text-video pairs where a single person appears, with duration from 4s to 16s. We additionally apply OpenPose (Cao et al., 2019) to only keep those with at least knee joints in the frame to avoid extreme close-ups. At the top of Fig. 10 we show some cases that we discard during the filtering process.

Note that interestingly, as we apply all the detection on middle frame, some earlier and later frames might not satisfy our requirements of full bodies. We choose to not specifically tackle these edge cases as they tend to have rich interactive contents with large-scale motions.

### B.2   HUMAN REMOVAL

To process the data, we take the first and last frames of a video for human removal to get the scene image.

**Human segmentation.** We apply GroundingDINO (Liu et al., 2023) with the keyword `human` to get bounding boxes for each human in the image. We apply SAM 2.1 with the bounding box as guidance to segment out the binary human mask.

**Inpainting.** We apply the SDXL diffusion inpainting model. To avoid fuzzy segmentation boundary, we use OpenCV to dilate each binary mask by 50 pixels so that it's guaranteed to cover the whole human area. The positive prompt we use is "natural, photorealistic, empty, environment, blank, background, bg", and the negative prompt is "person, human, text". For two people videos, we separate the two person masks, and does inpainting with each mask separately. At the bottom of Fig. 10 we show a few additional data samples, including mask and detected poses.

### B.3   PROMPT POST-PROCESSING.

We split the prompt by sentences. For each sentence, we ask the LLaMA model (Dubey et al., 2024) whether it describes the person or the background. If it's defined as a background prompt, we

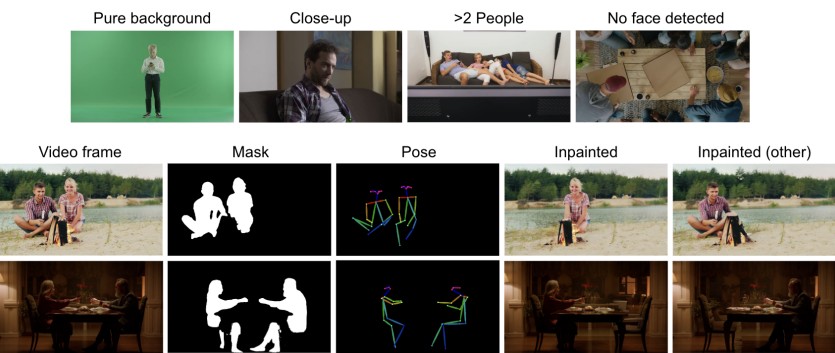

Figure 10: Additional illustration of our data processing pipeline. We include discarded data samples on top, and intermediate outputs of detection and filtering on bottom.

remove it from the caption. We additionally remove all sentences with the concept of camera in it, as we are not explicitly modeling any human-camera interaction.

## C IMPLEMENTATION DETAILS

### C.1 TRAINING

We use the base text-to-video model Movie Gen (Polyak et al., 2024) with 4B parameters, as described in Sec. 3. We train with landscape 256p, 16 frames per second, eight seconds per video. We fine-tune the full model with the text encoders frozen. We use a per GPU batch size of 1, and a learning rate of 1e-5. The training takes two days on 32 H100 GPUs.

### C.2 BASE MODEL

We explain some training details of our base model below. Refer to (Polyak et al., 2024) for more illustration. Note that while the training scheme and datasets are the same, we use a much smaller counterpart than the publicly announced Movie Gen model due to resource limitation.

We perform generation in a learned latent space representation of the video. This latent code is of shape $T \times C \times H \times W$. To prepare inputs for the Transformer backbone, the video latent code is 'patchified' using a 3D convolutional layer and then flattened to yield a 1D sequence. The 3D convolutional layer uses a kernel size of $k_\text{t} \times k_\text{h} \times k_\text{w}$ with a stride equal to the kernel size and projects it into the same dimensions as needed by the Transformer backbone. Thus, the total number of tokens input to the Transformer backbone is $THW/(k_t k_h k_w)$. We use $k_t = 1$ and $k_h = k_w = 2$, i.e., we produce $2 \times 2$ spatial patches.

We use a factorized learnable positional embedding to enable arbitrary size, aspect ratio, and video length. Absolute embeddings of $D$ dimensions can be denoted as a mapping $\phi(i) : [0, \text{maxLen}] \rightarrow \mathbb{R}^D$ where $i$ denotes the absolute index of the patch. We convert the 'patchified' tokens into separate embeddings $\phi_h, \phi_w$ and $\phi_t$ of spatial $h, w$, and temporal $t$ coordinates. We define $H_\text{max}, W_\text{max}$, and $T_\text{max}$ as the maximum sequence length for each dimension, which corresponds to the maximum spatial size and video length of the patchified inputs. We calculate the final positional embeddings by adding all the factorized positional embeddings together, and finally adding them to the input for all the Transformer layers.

### C.3 CONDITIONING BRANCH

We build our cross attention conditioning branch by concatenating the text and image features. Specifically, we apply 2 layers of text enhancer self attention, 2 layers of image enhancer deformable attention, then 6 layers of cross-attention with image as key/value and 6 layers of cross-attention with text as key/value. We combine the enhanced image feature with the pre-trained text feature for cross-attention with Transformer layer outputs.

## D    EVALUATION DETAILS

### D.1    BASELINE DETAILS

**T2I Inpainting.** We deploy a pre-trained text-to-image inpainting model on the given scene frame. We use the ground truth human bounding boxes from GroundingDINO's prediction as a guidance mask for inpainting. Because the baseline's text encoder is not designed for long prompts, we only take the first two sentences in our caption as the positive inpainting prompt. In practice, they are able to describe the human action and appearance adequately. Note that this is not an exactly fair comparison, as we give the model a ground truth bounding box. We are able to show that, however, our model is able to generate more natural interaction even without a pre-defined position signal.

**InstructPix2Pix and AnyV2V.** Both of them are based on InstructPix2Pix, except that the second one is an extension into video after editing the first frame. We use LLaMa (Dubey et al., 2024) to rewrite our prompts so that it falls into the instruction distribution. Instead of describing "the video shows a man", we rewrite the prompt as "adding a man". Similarly, due to the limit number of tokens the text encoder can take in, we only rewrite the first two sentences. We use the same prompt for both stages of AnyV2V.

Note that our baselines are mostly trained with squared images. Even though our model is exclusively trained with landscape videos, our Transformer architecture essentially enables generation of arbitrary aspect ratio. To accommodate the baselines, we use squared images for comparison in the main paper. We additionally provide some non-squared comparisons with the two image-based models in the next section.

### D.2    EVALUATION METRICS

**FVD.** FVD calculates the feature distance between two sets of videos. (the I3D features). We take the evaluation code and checkpoints from (Voleti et al., 2022). The metric is computed by

$$\text{FVD} = \|\mu_X - \mu_Y\|^2 + \text{Tr}\left(\Sigma_X + \Sigma_Y - 2\left(\Sigma_X \Sigma_Y\right)^{1/2}\right)$$

where $\mu_X$, $\mu_Y$ are the mean vectors and $\Sigma_X$, $\Sigma_Y$ are the covariance vectors.

**CLIP.** We compute the CLIP similarity between generated visual contents and the text prompts. For videos, the distance is computed every one second, and averaged across the whole video.

**Action Score.** We design this metric to eliminate the influence of human appearance and solely evaluate whether the inserted human is doing the correct action. We ask LLaVA-Next (Zhang et al., 2024) what the human is doing in a video, and provide samples of our action prompts as examples. We then compare the CLIP similarity between our prompt and the output. For the static images, we repeat the single static frame to make a video sequence. We notice that, as LLaVA is only taking a few key frames to answer the question, repeating the static frames is a reasonable way to decide human actions in an image.

### D.3    HUMAN EVALUATION DETAILS

We run a user study to recruit thirty-seven people evaluating the results of our model. We randomly shuffle the results of ours versus the three baselines and the three types of ablations. Among the users, fourteen fill out the small questionnaire with 10 groups of randomly selected results, and twenty-three of them fill out the complete questionnaire with 80 groups. People are asked to select their preference of the results based on four dimensions as described in the main paper.

## E    ADDITIONAL IMAGE BASELINE COMPARISON

In Fig 11, we show additional frame-wise comparisons with the image-editing baselines to demonstrate our model's superior ability. Note from the results how our model is able to keep the scene consistent instead of generating something semantically similar, and also able to insert a human without a mask.

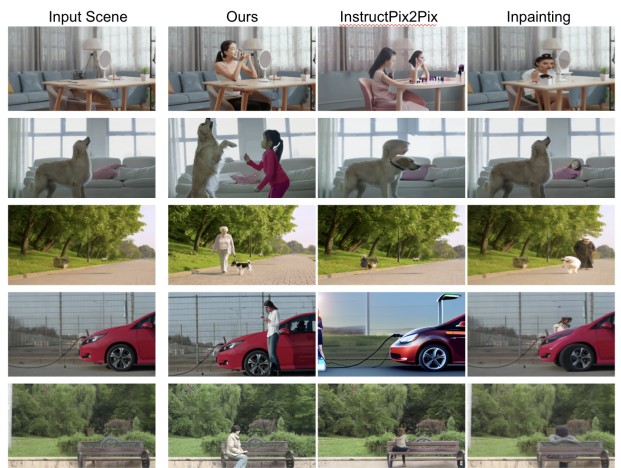

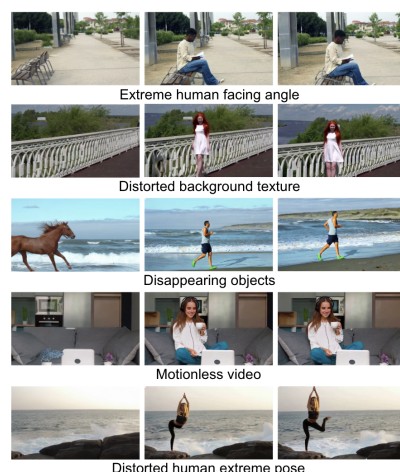

Figure 11: Additional comparison with baselines on non-square image inputs.

Figure 12: Limitation and failure cases of our model.

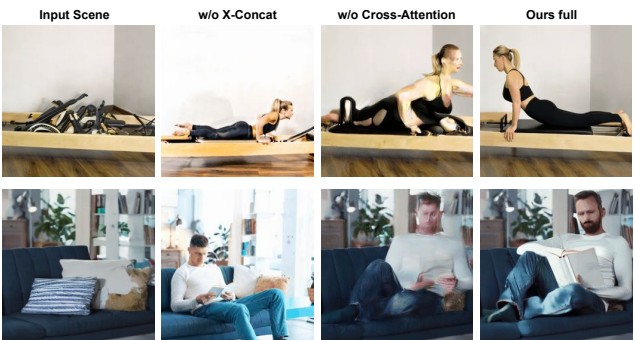

Figure 13: Comparison with alternative designs of our model.

## F  ABLATION VISUALIZATIONS

As shown in Fig. 13, our dual stream conditioning approach with both latent concatenation and feature enhanced cross-attention proves to be the best way of conditioning a T2V model on the scene image. Without latent concatenation, the model generates something semantically similar but not pixel-wise the same. Without fused cross-attention modules, the model is prone to generating distorted, unreasonable motions.

## G  LIMITATIONS

We discuss a few key limitations and failure cases we noticed in our current method, as shown in Fig. 12. Note that most of them are due to the base text-to-video model's limited capability, especially as we are basing our work on a smaller, lower resolution version. Overall, our method's quality greatly depends on the base model, and could be further improved with better model and more computing resources.

**Videos with limited motions.** Our model suffers from the common issue of generating videos with limited amount of motions (i.e. static videos). Specifically, we observe that some of our generated results have natural camera movements and environmental changes, while having the central character almost static. This is due to the data distribution which we use to train and fine-tune the model, and can likely be eliminated by providing higher quality fine-tuning dataset, or include motion guidance as an explicit condition to the model. Notably, we notice that our model is able to exhibit fair amount of motion with "action" prompts, like "running", "walking", "riking bike" whose

underlying semantic requires great movements. And results are more static with "status" prompts like "sitting", "lying", which merely describes an existing state. Regardless of the amount of motion, our model is always able to insert the person into the correct place with reasonable interaction.

**Human body distortion.** Similar to other text-to-video models, our model is not perfect in generating human movements, especially in examples with extreme human motion like doing sports. Specifically, we observe artifacts in limbs and hands when the model expects to generate fine-grained, large-scaled movements. We consider this a common issue of current text-to-video model, and could be improved by using better base model.

**Background texture distortion.** We notice that our model fails to keep scene consistent if there is complex geometry or texture in the input image. For example, architectures with repetitive structures, or periodic textures with fine details. This is also an on-going issue of state-of-the-art text-to-video models awaiting solution.

**Inpainting artifacts and object disappearing.** Our human removal inpainting algorithms fail on a few edge cases, where it removes the human but replaces it with an additional object. Training on these data teaches the model to sometimes "remove" existing objects in a scene and replacing it by a person, even if it shouldn't disappear in first place. We believe this is a relatively minor data quality issue and could be mitigated by using better inpainting off-the-shelf method, or add an additional round of data filtering.

**Extreme human facing angles.** We model is not able to generate back-facing human. This is due to how we filter the data: we detect faces and only keep those with the same face across the whole video, which in nature eliminates back facing videos. In cases where the inserted human is expected to face an extreme angle such that most of the faces are unseen from the camera, our model tends to insert person in a wrong direction.

## H    EXPLANATION OF LLM USAGE

We used LLMs only as general-purpose assistive tools. Specifically, after completing a full and meaningful draft of the paper, we used an LLM to polish the language for clarity and readability. In addition, we used an LLM to help search for related work by generating candidate paper lists. We did not use LLMs for research ideation, nor for writing the content or paragraphs of this paper.

