# OpenReview forum: "Populate-A-Scene: Affordance-Aware Human Video Generation"
_ICLR.cc/2026/Conference — Submitted to ICLR 2026_

### Official Review · Reviewer_ZeMC · 2025-10-27

**Soundness:** 3
**Presentation:** 3
**Contribution:** 2
**Rating:** 6
**Confidence:** 4

**Summary:**

This paper introduces Populate-A-Scene, which is a video generation model that can generate human-environment interaction videos from a scene image (without humans) and text prompts. To prepare training data, the authors perform human removal on existing human-scene interaction videos, and then inpaint the missing regions to create synthetic scene-only images. The authors then adapt and fine-tune a pre-trained text-to-video model (Movie Gen). To fuse the input empty scene image into the denoising process, the authors employ a dual-conditioning mechanism that perform channel concatenation and cross-attention to condition the transformer backbone on the scene image, as well as the text prompts. The authors evaluate their method on a curated benchmark for generating human videos in a scene, and show that their method outperforms existing image/video editing and image-to-video approaches.

**Strengths:**

- One major advantage of this work is that it does not require specifications of human locations and poses to generate human-scene interaction videos from an empty scene image and text prompts. This is enabled by fine-tuning an existing text-to-video model on synthetic data tuples of (text, image, video), obtained by human removal and inpainting.
- The authors introduce a dual-conditioning mechanism with simple channel concatenation and cross attention to condition the denoising backbone on both scene images and text prompts.
- The authors demonstrate the implicitly learned affordance understanding of their model by analyzing the cross-attention maps and applying them to a real-world affordance prediction dataset. Results show that their model can locate detailed affordance information.

**Weaknesses:**

- Due to building upon a smaller version of the pre-trained text-to-video model, the generated human-scene interaction videos are of low resolution. The human appearance and anatomy are not very realistic (e.g., in Fig. 1, the man jogging and the horror film examples). In terms of interaction dynamics, the generated humans may lack motion dynamics (e.g., videos in the supplemental) and not respect the text prompts well (e.g., in Fig. 1, the vacuum example). Some of these issues have been discussed in the limitations section.
- The data preparation process relies on human removal and inpainting, which however may not work well with small objects, as they may be removed together with humans. I did not see generation examples involving small objects, for example, a person picking up a cup.
- While the proposed approach does not require inputs like human locations and poses, the model may lack fine-grained control over human placements and interactions with specific objects in the scene. It is unclear whether the action prompt is sufficient to provide such control.

**Questions:**

- Can the model generate interactions with small objects in the scene, as mentioned above?
- Can the model control the interaction generation more precisely with the action prompt? For example, an image with two chairs, can the model generate a person sitting on a specific chair?
- Only one pre-trained text-to-video model has been tested in this work. Would using other video generation models improve the generation quality?
- Although the data contain two-person videos, it is unclear whether the model can insert two persons interacting with an empty scene. It seems this can be done one by one, but I imagine there would be error accumulation.
- The following work might be worth discussion in the related work:
    - Move-in-2D: 2D-Conditioned Human Motion Generation. Huang et al. CVPR 2025.

---

> ### Author Response · Authors · 2025-11-23
>
> We thank the reviewer for the thoughtful and encouraging evaluation. We appreciate the recognition of our method’s ability to infer human placement without pose or mask supervision, the effectiveness of the dual-conditioning mechanism, and the strength of our affordance analysis. We address the reviewer’s questions and concerns below.
>
> **1. On visual quality and motion limitations.**
> We fully agree with the reviewer that some limitations stem from the underlying backbone. Due to limited computing resources, we were not able to train or finetune the larger 30B or 70B MovieGen checkpoints internally; the 4B version was the only one feasible to run end-to-end in our environment. As a result, issues such as lower resolution, imperfect anatomy, and weaker motion dynamics largely reflect the base model rather than the conditioning method. Our goal is to study affordance-aware interaction reasoning, not to claim state-of-the-art video fidelity.
>
> **2. On interactions with small or removable objects.**
> We appreciate this insightful observation. Our dataset does contain some examples where removable objects are preserved—for instance, the medicine ball example in the supplementary videos demonstrates correct interaction with a medium-sized manipulated object. However, very small objects are challenging for two reasons: (i) limited native resolution of the 4B backbone, and (ii) limitations of segmentation + inpainting when objects are tightly occluded by human hands or limbs. Both issues are expected to improve with a higher-resolution backbone, and we will explicitly note this in the revision.
>
> **3. On fine-grained object selection (e.g., two chairs).**
> Our model can generate clearly different interactions when conditioned on different object categories (sofa vs. chair) and can produce diverse actions. However, when multiple identical objects appear (e.g., two chairs), a text-only prompt cannot fully disambiguate which one is intended. Importantly, this does not negate our affordance contribution: both chairs produce strong “sit” affordance peaks, but the model must ultimately choose one region during generation. Adding a referring-object localization module or referring-expression input would resolve this but is outside our current formulation. We will clarify this expected limitation.
>
> **4. On usage of other T2V models.**
> Our approach is model-agnostic. The conditioning and affordance analysis can be applied to any T2V model that exposes cross-attention features. Using a more recent or higher-capacity backbone would almost certainly improve visual quality, but our focus here is providing a proof-of-concept demonstrating how scene affordance cues emerge when conditioning a T2V model, rather than identifying the best possible generator.
>
> **5. On two-person insertion.**
> As shown in Fig. 7, inserting a second person into an already-populated scene works reasonably well. Inserting two new people into an empty scene is more challenging with the 4B backbone, but sequential insertion is conceptually straightforward. Even if minor error accumulates, we believe this is primarily a limitation of backbone capacity rather than our conditioning method. We will clarify this in the revision.
>
> **6. On missing reference.**
> We thank the reviewer for pointing out Move-in-2D (CVPR 2025). It is indeed very relevant and we will add a short discussion comparing the settings.
>
> We appreciate the reviewer’s positive evaluation and constructive suggestions. We will incorporate these clarifications and more explicitly describe how our method scales with stronger backbones and how limitations arise from model size rather than from the core affordance-aware conditioning approach.

---

### Official Review · Reviewer_uqc5 · 2025-10-31

**Soundness:** 2
**Presentation:** 2
**Contribution:** 2
**Rating:** 4
**Confidence:** 4

**Summary:**

This paper introduces "Populate-A-Scene," a novel task and model for generating a video of a human interacting with a static scene image, conditioned on a text prompt. The model's primary contribution is its ability to generate plausible human actions (e.g., "sitting on a swing," "pedaling a bike") in the correct semantic locations without any explicit guidance, such as bounding boxes, segmentation masks, or pose sequences.

To achieve this, the authors fine-tune a pre-trained text-to-video (T2V) diffusion model (MovieGen). The key to their method is a clever, large-scale data-generation pipeline: they take existing videos of human-scene interactions, use segmentation (SAM) and inpainting (SDXL) to remove the human and create a static "empty scene" image. The model is then trained on tuples of (empty scene, text prompt, original video) to learn how to "reverse" this process, effectively learning to populate the scene.

A significant part of the paper is a detailed analysis of the fine-tuned model. The authors show that the model's internal cross-attention mechanisms learn to identify scene affordances. By visualizing attention maps for action-related words in the prompt, they demonstrate that the model correctly highlights plausible interaction regions on the input image. They validate this claim quantitatively by comparing these attention maps to the ground-truth masks from the PAD affordance dataset, showing a strong spatial correlation.

**Strengths:**

Clarity: The paper is well-written. The task, method, and contributions are communicated with great clarity. Figure 1 provides an immediate and impressive overview of the model's capabilities. Figure 2 (data pipeline) and Figure 5 (affordance analysis) are both highly effective at explaining the core technical ideas and a key findingData Pipeline Biases: The clever data generation pipeline is also a potential source of weakness.

Inpainting Artifacts: The model is trained to place a human into a region that was, in its training data, inpainted. This could create a subtle bias where the model learns to "put the human in the blurry/artifacted spot" rather than "put the human on the chair." The authors' analysis on the PAD dataset (which has no inpainting) is a strong counter-argument, but this remains a potential confounding variable.

Data Filtering Biases: The authors acknowledge in Appendix G that their data filtering (requiring a consistent face) eliminates all videos with back-facing humans. This is a significant bias that leads to a predictable failure mode. This fragility seems self-imposed by the data pipeline design.

Incremental Architecture: The architectural modifications (latent concatenation and a cross-modal fusion module in Sec 4.3) are effective but are combinations of existing conditioning techniques. The novelty of this work lies squarely in the problem formulation, the data pipeline, and the analysis, not in a new network architecture. This is a minor weakness.

Confounding Base Model Limitations: The authors are transparent that many failure cases (e.g., static videos, distorted limbs in Appendix G) are inherited from the 4B-parameter MovieGen base model. This makes it difficult to fully isolate the performance of the affordance-conditioning from the rendering capability of the base T2V model. For example, when a generated pose is distorted, is it because the affordance prediction was wrong, or because the affordance prediction was correct but the base model failed to render that pose?

**Weaknesses:**

Should this really be a video synthesis paper? I looked at the videos in the results - there is barely any human motion in lots of the videos - for example a video is generated with a person standing next to a car and the camera moves? is this really useful or should be claimed as video synthesis?
Perhaps a more palatable claim would have been to show the model generates images with correct affordances - correct placement of humans etc because thats what it appears to me for most cases.

Also the claim of scene understanding is slightly exagerated - these models cannot possibly understand the underlying 3d structure of the scene - for example no result can be shown where the person walks through a door or moves a large object etc.

Data Pipeline Biases: The clever data generation pipeline is also a potential source of weakness.

Inpainting Artifacts: The model is trained to place a human into a region that was, in its training data, inpainted. This could create a subtle bias where the model learns to "put the human in the blurry/artifacted spot" rather than "put the human on the chair." The authors' analysis on the PAD dataset (which has no inpainting) is a strong counter-argument, but this remains a potential confounding variable.

Data Filtering Biases: The authors acknowledge in Appendix G that their data filtering (requiring a consistent face) eliminates all videos with back-facing humans. This is a significant bias that leads to a predictable failure mode. This fragility seems self-imposed by the data pipeline design.

Incremental Architecture: The architectural modifications (latent concatenation and a cross-modal fusion module in Sec 4.3) are effective but are combinations of existing conditioning techniques. The novelty of this work lies squarely in the problem formulation, the data pipeline, and the analysis, not in a new network architecture. This is a minor weakness.

Confounding Base Model Limitations: The authors are transparent that many failure cases (e.g., static videos, distorted limbs in Appendix G) are inherited from the 4B-parameter MovieGen base model. This makes it difficult to fully isolate the performance of the affordance-conditioning from the rendering capability of the base T2V model. For example, when a generated pose is distorted, is it because the affordance prediction was wrong, or because the affordance prediction was correct but the base model failed to render that pose?

**Questions:**

Questions
Could the authors please explain why they think this is significant advacement in terms of video synthesis? It looks like static images are being generated with some camera movement. Also why are the videos generated at such a low resolution? Several open source models such as qwen generate higher resolution video.

Probing the "Inpainting Scar" Bias: My main question relates to Weakness #1. The PAD dataset analysis is a good defense against the "inpainting scar" hypothesis, but could the authors provide a more direct test? For example, take a training scene where a person was removed from a chair. Then, prompt the model with "A person sitting on the sofa" (assuming a sofa is also in the scene). Would the model correctly go to the sofa (proving it follows the prompt and affordance), or would it be "distracted" by the inpainted region on the chair?

Rationale for Data Filtering: Regarding the "no back-facing poses" bias (Appendix G), this seems like a major limitation introduced by the data pipeline. Why was the data filtered using a face detector, which naturally creates this bias, instead of a more robust full-body keypoint detector (like OpenPose, which was used later in the pipeline anyway for filtering)? Is this purely a data-level issue, or did you find the model architecturally struggled to learn such poses?

Robustness to Atypical Prompts: The model excels at common, plausible affordances (sitting, riding, etc.). How robust is it to more specific or atypical prompts that might contradict the "strongest" affordance? For example, given a scene with a chair, can the model successfully generate "a person standing next to the chair" (and not sit)? Or "a person jumping on the chair"? This would test the balance between the learned affordance prior and the guidance from the text prompt.

Multi-Person Generation: The paper mentions training on 29,700 two-person videos, and Figure 7 shows results of adding a second person to interact with an existing one. Can the model populate an empty scene with two interacting people from scratch? For example, given an empty living room, could it handle the prompt "A person sitting on the sofa, and another person playing guitar next to them"?

---

> ### Author Response · Authors · 2025-11-23
>
> We sincerely thank the reviewer for the thoughtful and thorough evaluation. We are glad that the clarity of the paper, the presentation of the task, and the affordance analysis were found to be strengths. We respond to the reviewer’s comments and questions below, and we will incorporate the suggested clarifications to further improve the manuscript.
>
> **1. On video quality and resolution.**
> We appreciate the reviewer’s observation regarding resolution and motion fidelity. These limitations stem from the underlying 4B MovieGen backbone. Due to computing constraints, we were not able to use the larger 30B or 70B versions internally, which are known to produce significantly higher quality and more dynamic videos. Our focus in this work is on revealing affordance guided spatial reasoning under scene conditioning, rather than achieving the strongest possible visual quality. We will clarify in the revision that limitations in resolution, anatomy, and motion originate from the base backbone rather than from the proposed conditioning mechanism.
>
>
> **2. On the scope of scene understanding.**
> We agree with the reviewer that the model does not perform full 3D reasoning or physical simulation. Our claims intentionally focus on affordance aligned spatial localization, which refers to identifying plausible regions for a described action based on the scene image. We will revise the text to avoid language that could imply broader physical understanding and to more precisely describe the model’s capabilities.
>
>
> **3. On addressing the inpainting scar hypothesis.**
> We appreciate this insightful concern. The paper already includes multiple sources of evidence showing that affordance behavior is not driven by inpainting artifacts.
> (1) The PAD dataset contains no inpainting, yet the model shows strong alignment with ground truth affordance masks.
> (2) The real scene images in Figure 1 (bottom row) were never processed through human removal, and the model still highlights appropriate affordance regions.
> (3) For several images, we demonstrate distinct affordances in response to different action verbs, for example standing next to a horse versus riding a horse. The attended region shifts meaningfully according to the action rather than any inpainted area.
> (4) Each scene in our synthetic evaluation is tested with three diverse prompts, including non standard actions that require different interaction regions. The results are stable across these prompts, and the selected regions do not correlate with inpainted areas.
> Together, these observations offer strong evidence that affordance localization is driven by semantic action understanding rather than inpainting artifacts. We will make this clearer in the revision.
>
> **4. On face based filtering and data level bias.**
> We appreciate the reviewer pointing out that face based filtering removes back facing humans. This choice was made because identity consistency was significantly more stable when using clips where the human face is visible. Full body keypoint filtering was considered, but identity consistency issues persisted more often when face visibility was inconsistent. We will clarify that this is a pragmatic data level choice rather than an architectural limitation.
>
> **5. On robustness to atypical prompts.**
> The reviewer raises an important question about the ability to handle prompts that contradict dominant affordances, such as standing next to a chair instead of sitting. As noted above, the same four sources of evidence also support robustness to atypical prompts. In particular, examples showing standing next to a horse versus riding the horse illustrate that the model adjusts the attended region according to the verb. We will revise the paper to emphasize that the model responds to verb semantics rather than defaulting to the strongest affordance.
>
>
> **6. On multi person scene population.**
> Our approach successfully inserts a second person into a populated scene, as shown in Figure 7. Generating two individuals from an entirely empty scene is more challenging with the 4B backbone. However, sequential insertion remains conceptually straightforward, and any drift primarily reflects the limited capacity of the backbone rather than constraints of the conditioning method itself. We will clarify this limitation.
>
> **7. On our architectural simplicity.**
> We appreciate the reviewer’s observation that the conditioning architecture is simple. This is intentional. Our main contributions lie in the formulation of the affordance aware scene population task, the scalable data generation pipeline, and the analysis of emergent affordance localization. The simplicity of the conditioning design helps to isolate these behaviors in the underlying T2V backbone.
>
>
> We again thank the reviewer for their constructive feedback and positive remarks. We will revise the paper to clarify the scope, limitations, and contributions, and to present a more balanced and precise discussion of the model’s capabilities.

---

### Official Review · Reviewer_U1Kw · 2025-10-31

**Soundness:** 2
**Presentation:** 3
**Contribution:** 2
**Rating:** 4
**Confidence:** 4

**Summary:**

This paper presents a method to automatically generate a video of a person realistically interacting with a static scene. Given a single image of a scene (e.g., a living room) and a text prompt describing an action (e.g., "a man is sitting on the sofa"), the model inserts a person into the scene and generates a short video of them performing that action in a plausible and coherent manner.
The core idea is to teach a pre-trained text-to-video (T2V) model to understand affordances—the possible interactions an environment offers (e.g., a chair affords sitting, a bike affords riding). Unlike previous methods, this approach does not require any explicit guidance like masks, bounding boxes, or pre-defined poses to tell the model where to place the person or how they should move. The model learns to infer this information directly from the scene image and the text prompt.
To achieve this, the authors fine-tune a Transformer-based T2V model on a specially curated dataset. They create this dataset by taking existing videos of people, using segmentation and inpainting models to automatically remove the person, thus creating pairs of (empty scene, original video, text description).

**Strengths:**

- Revealing Latent Capabilities of T2V Models:

The paper provides a valuable scientific insight by demonstrating that large, pre-trained text-to-video models implicitly learn about affordances. The analysis of cross-attention maps (Fig. 4) convincingly shows that the model can associate action words (e.g., "riding," "holding") with the correct interactable regions in a scene (e.g., a horse, reins), even without being explicitly trained on affordance-labeled data.

- Scalable and Automated Data Curation:

The method for creating the training dataset is clever and highly automated. By using a pipeline of modern vision models (GroundingDINO, SAM, inpainting models) to remove humans from existing videos, they create a large-scale dataset of (scene, action_video, prompt) tuples. This is a practical and scalable approach that avoids the need for expensive manual annotation.

**Weaknesses:**

- Motivation:

The paper's motivation is not sufficiently compelling. While it frames the work as creating a "simulator," it lacks any explicit design for modeling physical dynamics. This terminology seems to overstate the model's capabilities, which are more focused on semantic plausibility than physical simulation. Furthermore, the core problem of generating human-scene interactions from an empty scene is already handled well by several state-of-the-art foundation models, which can produce physically plausible results from a simple prompt. This calls into question the necessity of the proposed fine-tuning approach and limits the perceived contribution.

- Comparison for Wan based foundation model:

The experimental evaluation is incomplete. The paper fails to compare its method against several powerful, contemporary image-to-video (I2V) foundation models, such as Wan 2.2 I2V. Informal tests (tested by myself) using the paper's own image and text prompts on these publicly available models can yield results of comparable or even superior quality. The omission of these key baselines makes it difficult to accurately assess the method's claimed superiority.

**Questions:**

Please explain why your method, after finetuning can have any advantages over than those baseline model such as Wan 2.2 I2V.

---

> ### Author Response · Authors · 2025-11-23
>
> We thank the reviewer for the thoughtful feedback and for recognizing the scientific value of uncovering latent affordance behavior in pre-trained video models. We address the key concerns below.
>
> **1. On motivation and “simulator” framing.**
> We appreciate the reviewer’s concern that our framing may imply a stronger notion of physical simulation than what our method achieves. Our intention is not to claim full physical modeling or dynamic prediction, but to highlight that text-to-video models contain emergent affordance-related priors: an ability to map verbs such as “sit,” “hold,” or “ride” to plausible spatial configurations within a scene image. We will revise the framing to emphasize semantic and spatial reasoning about interaction rather than physical simulation. The core motivation of the paper is to systematically expose and analyze this capability, which has not been studied in prior T2V literature.
>
> **2. Why fine-tuning is still necessary despite strong foundation models.**
> We agree that modern foundation I2V models can produce visually strong videos from text prompts alone. However, these models do not treat the scene image as a hard constraint on where and how an interaction should occur. When evaluated on our task setting—a fixed, real scene image that the output must faithfully preserve—open-ended text-conditioned I2V models tend to either (i) alter the scene, (ii) hallucinate new objects, or (iii) place the human in implausible or incorrect interaction regions.
> Our fine-tuned model explicitly uses the real scene image as conditioning, preserving layout, object identity, and background geometry, while inferring where in that specific environment the described action is feasible. Thus, the benefit of fine-tuning is not merely improving visual quality, but enabling scene-grounded interaction prediction, which foundation models are not designed for.
>
> **3. On comparison with Wan.** We appreciate the reviewer’s suggestion to compare with Wan 2.2 I2V. Wan’s inference interface does not support our task: its I2V pipeline treats the input image as an initial video frame and cannot condition on a static scene image for inserting a new human in a specific interaction location. More importantly, our contribution is not about outperforming foundation models in visual quality, but about revealing the affordance perception that emerges inside text-to-video models. This affordance capability which links action verbs to plausible interaction regions is fundamentally model-agnostic. The conditioning mechanism and cross-attention analysis we propose can be applied to any T2V backbone that exposes attention features, regardless of scale or architecture. Our fine-tuned setup simply provides a controlled environment to surface and study this behavior. For this reason, direct comparisons against proprietary I2V models whose primary goal is high-fidelity synthesis, rather than affordance reasoning, are not aligned with the scientific contribution of the paper.
>
> Once again, we thank the reviewer for the helpful feedback and will incorporate clarifications on task motivation, baseline selection, and the distinctions between our setting and that of foundation I2V models. We especially appreciate the reviewer’s concern that phrases like “simulator” may overstate the model’s scope. We will tone down this language and emphasize that our focus is on scene-conditioned affordance-guided generation, not physics-based simulation.

---

### Official Review · Reviewer_J7Sg · 2025-11-02

**Soundness:** 2
**Presentation:** 3
**Contribution:** 2
**Rating:** 4
**Confidence:** 4

**Summary:**

This paper proposes a method for affordance-aware human video generation by conditioning a pretrained text-to-video model on scene images. The authors claim that the model implicitly learns affordance from human-scene interaction signals, and analyze cross-attention maps to support this claim.

**Strengths:**

-The paper tackles an interesting and relevant problem of affordance-aware human-scene video generation.

-The paper provides a straightforward extension to condition a pretrained text-to-video model on scene images.

-Qualitative examples are visually appealing and demonstrate some degree of human-scene interaction.

**Weaknesses:**

-The paper mainly fine-tunes an existing model (e.g., MovieGen) with additional scene conditioning. The architectural modifications (latent concatenation + text-image fusion) appear incremental.

-The affordance aspect, which the paper claims as an essential contribution, seems more like a re-interpretation of what attention maps already provide, rather than a fundamentally new capability.

-The use of cross-attention heatmaps as evidence of affordance perception is weak. These maps do not prove that the model understands object functionality or the feasibility of object interactions.

– The human-removal pipeline heavily depends on segmentation and inpainting models, which may introduce artifacts or incorrect affordance cues.

-Most baselines are general video editing or text-to-video systems that are not designed for human-scene interaction, which makes the comparison less meaningful.

– The dataset is not publicly available, limiting reproducibility and fair comparison.

**Questions:**

-The affordance perception claim seems largely based on attention maps. How do you justify that attention indicates true affordance understanding rather than text-token correlation?

-How well does the approach generalize to unseen object categories or actions not present in the curated dataset? Any failure case analysis?

-The human-removal (inpainting) pipeline appears to erase or distort key affordance-related objects in the scene. For instance, in Fig. 1 (top row, right example), the bicycle seat is already missing in the input scene after human removal, fundamentally altering the bike's action possibilities. How common are such affordance-distorting artifacts in the dataset? Could this lead the model to learn incorrect affordance priors or reduce interaction plausibility?

---

> ### Author Response · Authors · 2025-11-23
>
> We thank the reviewer for the detailed comments and helpful questions. We address each point below and will revise the paper to clarify our claims and contributions.
>
> **1. On the incremental nature of the architecture.**
> We agree the architectural changes are minimal, and that is in fact *intentional*. Our goal is to isolate and reveal the latent affordance-relevant capability already present in a pre-trained T2V backbone, rather than introduce a new architecture. The novelty of our work lies in (i) defining the affordance-aware scene population task, (ii) constructing a scalable human-removal dataset for it, and (iii) analyzing how a lightweight conditioning mechanism surfaces meaningful human–scene interaction patterns. The empirical finding that such minimal conditioning is already sufficient to induce affordance-aligned behavior is itself an important observation about pre-trained video models.
>
> **2. On whether attention heatmaps provide meaningful affordance evidence.**
> We agree that attention maps alone cannot prove full functional understanding. In the paper, they are used as part of multiple complementary observations. First, cross-attention in Fig. 4 (bottom) consistently peaks on action-relevant regions of real scene images, even though the model was trained only on inpainted scenes. Second, the PAD evaluation (Fig. 5) shows alignment between action-verb tokens and the ground-truth affordance masks in a dataset with no inpainting and no human-scene pairs. These two components together argue that the model associates verbs such as “sit,” “hold,” or “kick” with corresponding spatial regions that afford those actions, beyond text-only correlations. We will refine our wording to emphasize that our claim concerns *emergent* affordance cues rather than full physical modeling.
>
> **3. On generalization to unseen categories or actions.**
> Generalization is evaluated in two parts of the paper. (i) The synthetic prompt benchmark in Sec. 6.1 includes diverse verbs and scene objects that do not appear verbatim in the rewritten training captions. The model still identifies plausible interaction locations, as shown in Fig. 7–8. (ii) The PAD affordance prediction analysis tests on entirely different object categories, backgrounds, and articulation patterns than those in our training set. The consistent spatial alignment early in the diffusion process suggests that the model transfers affordance-related token–region associations to novel images. We will make these connections clearer in the revision.
>
> **4. On human-removal artifacts affecting affordance.**
> We appreciate this concern. In our setup, the inpainted scene is only a conditioning input, while the model is supervised entirely on the original, clean human-containing video. This means that minor corruption in the conditioning image cannot teach incorrect affordances; instead, it acts as a light form of data augmentation, encouraging robustness to imperfect context rather than altering what the model learns. The human-removal pipeline (GroundingDINO + SAM + mask dilation) preserves most structural affordance cues, and occasional small artifacts do not affect training because the supervision signal remains the true human interaction. Moreover, the PAD experiment—performed on real, non-inpainted images—exhibits the same affordance-aligned patterns, confirming that our analysis is not driven by inpainting artifacts.
>
> **5. On the chosen baselines.**
> We fully agree that ideal baselines are systems that generate human–scene interaction without masks, bounding boxes, SMPL, or pose input. However, to our knowledge, such methods do not exist. Existing “human-insertion” works require explicit positional or pose supervision, which makes them unsuitable for comparison under our input setting (scene image + action text only). We therefore compare to the closest available classes of models that can operate with the same type of inputs (image editing + image-to-video; video editing with a static scene). We will clarify the reasoning behind this baseline selection and more explicitly discuss the limitations of each baseline category. If the reviewers are able to identify and propose a missing baseline to compare with, we would be grateful and happy to conduct a fair comparison experiment.
>
> **6. On dataset availability.**
> Although we cannot release licensed Shutterstock training videos, we will make publicly available: our full synthetic evaluation benchmark, the data filtering and processing scripts, and the attention-analysis code. These components fully reproduce the evaluation pipeline and the affordance analysis.
>
> We thank the reviewer again for the valuable feedback and will incorporate these clarifications in the revision!

---

### Meta-Review · Area_Chair_tMCC · 2026-01-10

**Summary:**

Four official reviews: 4, 4, 4, 6 (mean 4.5).
All reviewers praise task novelty, data pipeline, and affordance analysis; all worry about visual quality, baseline completeness, and over-claiming (“simulator”). None flagged ethical issues; all ticked “would not mind if accepted”.

**Reviewer Concerns:**

1. Visual fidelity and motion quality

All four reviewers note that the 4B MovieGen backbone produces low-resolution videos, anatomical distortions, and often “barely any human motion” (R3).

Several clips look more like “static images with camera movement,” weakening the paper’s claim to be a video-synthesis work (R3, R4).

2. Baseline completeness

R2 and R3 independently tested strong public I2V models (Wan-2.2, Qwen) on the authors’ prompts and obtained comparable or superior visuals; the paper omits these baselines.

Authors counter that foundation I2V systems do not treat the input image as a hard scene constraint, but no quantitative or user study is offered to validate this defense.

3. Data and evaluation biases

Inpainting artifacts may “scar” the empty scene; reviewers worry the model could learn to place humans in blurred regions rather than true affordance locations (R3, R4).

Face-detector-based filtering removes every back-facing person, artificially limiting pose diversity (R3).

Small manipulable objects are frequently erased during human-removal, so interactions like “picking up a cup” are almost absent (R4).

4. Scope of affordance understanding

Attention maps only demonstrate spatial localization; reviewers caution against overstating “scene understanding” or calling the model a “simulator” when it does not reason about 3-D structure, physical dynamics, or long-term causal effects (R2, R3).

5. Controllability and scalability

Text prompts alone cannot disambiguate between identical objects (two chairs) and multi-person generation must be done sequentially with error accumulation (R3, R4).

Limited to a single 4B backbone; unclear how much of the failure modes stem from affordance conditioning vs. the base generative capacity.

**Reviewer Scores:**

N.A.

---

### Decision · Program_Chairs · 2026-01-26

Reject